# The Road to Healthy Ageing: What Has Indonesia Achieved So Far?

**DOI:** 10.3390/nu13103441

**Published:** 2021-09-28

**Authors:** Ray Wagiu Basrowi, Endang Mariani Rahayu, Levina Chandra Khoe, Erika Wasito, Tonny Sundjaya

**Affiliations:** 1Occupational Medicine Division, Department of Community Medicine, Faculty of Medicine, Universitas Indonesia, Jakarta 10430, Indonesia; levina.chandra01@ui.ac.id; 2Medical and Science Affairs Division, Danone Specialized Nutrition Indonesia, Jakarta 12950, Indonesia; erika.wasito@danone.com (E.W.); tonny.sundjaya@danone.com (T.S.); 3Department of Anthropology, Faculty of Social and Political Science, Universitas Indonesia, Depok 16424, Indonesia; endang.cultural.psychology@gmail.com; 4Laboratorium Political Psychology, Faculty of Psychology, Universitas Indonesia, Depok 16424, Indonesia

**Keywords:** healthy ageing, aging population, nutritional status, demographic, health profile

## Abstract

The World Health Organization (WHO) has projected that the world should prepare for an aging society. As the fourth most populous country in the world, the elderly population in Indonesia is also continuously growing. In 2010, the proportion of the elderly group was merely 5%, and it is expected to increase to 11% in 2035. Understanding the current situation of the adult population in Indonesia would be crucial to prepare for the future aging population. This article analyzed the current socio-demographic status, nutrition status, nutrient intake, and health profile of the current Indonesian adult population through a literature review. The key issues to prepare for healthy aging in Indonesia are summarized. Acknowledging the profile of the adult and senior adult population in Indonesia will provide beneficial information for all stakeholders in preparing Indonesia for a better healthy aging population with improved quality of life.

## 1. Introduction

As the fourth most populous country in the world, Indonesia is currently entering the demographic dividend period, where the proportion of the working-age population is on the rise and outweighs the group of dependents, that is, the under 15 and the above 65 age group. In 2019, the number of people of productive age (15- to 59-year-olds) was 172.7 million, while the senior adult group (60-year-olds and above) was merely 10% (26.8 million) of the total population [1]. With the vast amount of productive population, Indonesia would have a greater responsibility to ensure a healthy aging population in the future.

Not only is there an increase in the productive age group but also the elderly population is also continuously growing. In 2010, the proportion of the elderly group was merely 5%, and it is expected to increase to 11% in 2035. The demography of the population would shift from high birth rate to low birth rate and high death rate to low death rate. Data from BPS-Statistics Indonesia show that the life expectancy in Indonesia is increasing, from 72.51 in 2015 up to 75.47 in 2045 [2].

The World Health Organization (WHO) has projected that the world should prepare for an aging society. Globally, it is estimated that the number of people aged 60 years and older would reach 2 billion in 2050, doubled from 1 billion in 2019 [3]. In addition, 80% of all older people will live in low- and middle-income countries [4]. As people live longer, they will be posed with various challenges in terms of health and quality of life. Non-communicable diseases (NCDs) are becoming a big threat to individuals and will continue to do in the long term. In 2018, WHO estimates that NCDs kill 41 million people each year (71% of all deaths globally) [5]. Cardiovascular diseases, cancers, respiratory diseases, and diabetes account for over 80% of all premature NCD deaths [5]. These diseases are also linked with disability, dependency, and long-term care needs [6]. An analysis in 23 low- and middle-income countries estimated that the economic losses from three non-communicable diseases (heart disease, stroke, and diabetes) would reach USD 83 billion between 2006 and 2015 [7].

Despite the potential challenges and vulnerability among the elderly population, we should also acknowledge the opportunities that arise from this group. These older persons live longer and have accumulated expertise, knowledge, and experience far better than the younger population. Many countries seem to fail to recognize these and assume the elderly population to be a burden rather than an asset to improve the country’s development.

As an inevitable part of the lifecycle, it is impossible to prevent aging in the population. Hence, a healthy aging population needs to be prepared as early as possible. If the health of the productive population can be maintained and enhanced, the demographic bonus can be utilized maximally to improve Indonesia’s development. Understanding the current situation of the adult population in Indonesia would be important to prepare for the future aging population. Therefore, this paper aims to provide the socio-demographic status, nutrition status, nutrient intake, and health profile of the current Indonesian adult population and to select the opportunities for improving their quality of life. The summary of the issues to be overcome can be seen in Table 1.

## 2. Methods

This paper analyzed the current nutrition and health status through a literature review. We identified articles through multiple channels, particularly official documents and reports published from national bodies, and also regional conferences. We also included literature from non-governmental organizations that provides data on the national level. Articles were either in English and/or Indonesian language, and limited to 10 years of publication time. The study design was not limited and may include trials, surveys, and observational studies.

## 3. Results

### 3.1. Socio-Demographic Status

Indonesia is currently in a bonus demography period with a larger productive age population than the nonproductive age. The dependency ratio is <50% as a result of the bonus demography would end around 2036. This period provides the opportunity for socio-economic growth for Indonesia. Nevertheless, we should be aware of the increase in the old-age dependency ratio, that is, the ratio of the elderly (60+) compared to the productive age (15–59), which keeps increasing over the years. The Central Agency on Statistics published a population projection based on the national census of the population. The elderly population (age 60+) would grow from 10.1% in 2020 to 18.0% in 2040, with the female proportion higher than male. This trend would be followed by increasing life expectancy and a lower fertility rate. The life expectancy would increase from 72 (2020) to 76 years (2045), while the fertility rate would fall from 2.28 (2020) to 1.94 (2045) births per woman [1,8]. In terms of geographical distribution, Java Island, which has the highest population density in Indonesia, also ranks as having the highest elderly population, with three top provinces: Yogyakarta, Central Java, and East Java. Previously, more elderly people resided in rural areas; however, recently, the elderly in the urban areas have overtaken those in rural areas (52.8% vs. 47.2%) [1].

In regards to literacy rate, compared to other age groups, the elderly have the lowest literacy rate (<92%), especially those living in rural areas (81.3%). While for the adult group aged 35–49 years old, the literacy rate was relatively high, that is, over 95% both in urban and rural areas [1]. This could have been influenced by government support in providing non-formal education to the adult population and giving financial support to those with low socioeconomic status. Data in 2016–2019 indicated that the number of senior adults with secondary school, high school, and university education levels is increasing over the years [9,10].

With a higher literacy rate and better education level in the adult population, there is no doubt that the adults have the highest proportion of internet access, with the peak at the age group of 25–49 years. The elderly group is also becoming more adapted to the use of technology. Even though, the senior adults were less likely to use technology in general, compared to the younger population [11]. In Indonesia, the proportion of senior adults using handphones and accessing the internet grew significantly between 2015 and 2019, that is, 23.8% to 43.1% and 1.6% to 7.9%. Their main objectives behind using the internet were social media and messaging, and also getting information, whether news or entertainment. About 80.3% of pre-senior adults used the internet for social media, and merely 58% for messaging [12]. Nonetheless, the elderly are more prone to share hoax news or unproven information. There were possible arguments supporting this evidence such as the elderly lacking the required digital literacy skills compared to the younger generation and the risk of cognitive decline increasing their likelihood to believe in fake news [13].

As commonly seen in Asian culture, most senior adults live in a three-generation house or with their families [9]. Nursing homes are relatively unpopular among Indonesian residents, and merely 12–15% of neglected senior adults in Jakarta stay in nursing homes [14]. Indonesia has a strong culture in which children should take care of their elderly parents. Therefore, it is quite apparent that the source of income is mostly the work of a household member or transfers from their families (93.5%), and only 5.8% receive incomes from their retirement pay-out [9]. In terms of economic status, about 40% are in the middle class, and 20% are at the top level. The middle and upper class has a better awareness of nutrition and supplements than the lower class. Meat, fruits, prepared food and beverage, seafood, and dairy products are consumed in significantly higher amounts by people in the top 20% compared to the bottom 20%. While for non-food items, the main expenditure is housing and household facilities [15].

### 3.2. Healthy Aging Concept

Generally, it is well accepted that the cut-off of the senior adult group starts at the age of 60 [16]. The current official retirement age in the private sector is 57 years old. Nevertheless, this is not strictly enforced, and the retirement age ranges from 56 to 65 years old. The Indonesian Ministry of Health made a more detailed classification proposing a pre-senior age group, that is, ages 45 to 60 years old, to be the target for early healthcare, NCD prevention, and intervention before entering the senior stage [9,17].

The World Report on Aging and Health as published by the WHO provided a framework for the healthy aging concept [11]. WHO defines healthy aging as “the process of developing and maintaining the functional ability that enables wellbeing in older age” [18]. The term functional ability refers to the ability of older people to meet their basic needs, to learn and grow and make decisions, to be mobile, to build and maintain relationships, and to contribute to society. It is a combination of intrinsic capacity (i.e., all the mental and physical capacities) and environment characteristics (i.e., the extrinsic factors surrounding the individual, such as people and their relationships, attitudes and values, health and social policies). This healthy aging concept emphasizes the need for inter-sectoral action and supporting elderly people as a resource for their families and communities.

In Indonesia, the Ministry of Health uses the continuum of care approach, by offering an integrative care system starting from conception to the elderly stage. Various health programs are managed through the complete lifestyle, targeting a quality aging population. The quality dimensions include measures of health (i.e., absence of disease), ability to perform independently, being socially active, and ability to contribute to themselves and the surrounding communities [9]. Besides the continuum of care approach, the National Family Planning Coordinating Agency offered the concept of the robust elderly through the elderly family education program in order for them to become healthy (physical, social, mental), independent, active, productive, and have the optimal quality of life [19].

The concept of healthy aging is also known to be linked to active participation in society. Adult and senior adult populations have high social activity, especially due to attending funerals and religious and volunteering events [20]. They are accustomed to attending many traditional weddings, funerals, or visiting family and friends. Travelling is mainly done for religious purposes. As the largest Muslim population in the world, pilgrimages are the major traveling activity performed by senior adults. About 62% of pilgrims are aged above 50 [20,21].

#### 3.2.1. Nutritional Status

From the national basic health survey, in adults through to the elderly, the prevalence of being underweight decreased from 2013 to 2018 across all age groups [22,23]. Underweight prevalence in 2018 was about 10% in young adults, decreased in middle-aged adults, increased again to about 10% at ages 60 to 64 years, and increased to 21% at ages above 65 years [23]. The highest prevalence of under-nutrition is found in the elderly over 65 years. About one out of five elderly people were underweight. Among Women of Reproductive Age (WRA) under-weight prevalence decreased across time from 2013 to 2018 and across age groups [22,23].

As it is also well known, the other side of the nutrition condition also existed among the Indonesian adult population. From the basic national health research, the prevalence of being overweight increased from young adults to the 45–49-year age group, then decreased again in older age groups from 12% to 17%, then to 10% [23]. About one to two persons out of 10 are overweight. The prevalence of being overweight is higher among females across age groups, with the difference being between 3 to 4 percentage points and increased from young adults to those aged 40–44 years, then decreased in older age groups [23]. The prevalence of central obesity increased across time (2013 to 2018) for all age groups [22,23]. This prevalence increased in the adult population with the age range from age 25 to 54 years and decreased again afterward [23], with the highest prevalence of central obesity being among those aged 45–54 years (42%) and a higher prevalence in urban areas.

Based on IFLS-1 to 4 data (1993–2014), the prevalence of overweightedness and obesity among young adults both male and female has also shown an increasing tendency [24]. In 2007, Prevalence Ratio (PR) overweight = 1.76 (95% CI: 1.64–1.89); PR obesity = 3.00 (95% CI: 2.46–3.69), and in 2014, PR overweight = 2.26 (95% CI: 1.97–2.60); PR obesity = 4.73 (95% CI: 3.87–5.78). Over this 15-year period, the proportion of overweight men and women with a BMI ≥ 25 almost doubled [25].

#### 3.2.2. Nutrient Intake

In 1952, the government introduced the slogan “4 sehat 5 sempurna (4 healthy 5 perfect)” and made a successful campaign for dietary guidelines. However, this slogan carried a misperception of the message to the public by putting a huge emphasis on milk consumption. In 1995, Indonesia changed the recommendation to balanced nutrition guidelines. Unfortunately, the campaign for balanced nutrition was not as successful as the previous one. Despite the poor response from the public, the government is trying to improve the campaign method and launched “on my plate” guidelines for every meal. The government also published recommended dietary allowance (RDA) guidelines and updates them every 5 to 10 years. The recent RDA in 2019 stated the need for a higher intake of protein, vitamin E, and calcium for adults and senior adults. In age groups above 50, the intake of 200 g milk was recommended. Sadly, experts have pointed out that the public perceives milk as a nutrient only for children, and there is no habit of drinking milk for adults and senior adults [26,27].

A national survey revealed that adults and senior adults had a low intake of fruits and vegetables and a high intake of foods that possess health risks [22,23]. The average portion intake for fruit was 0.7 per day and vegetables was 1.3 per day in the adult population. Comparing to other age groups, this rate was similar as only around 5% of the general population had a sufficient intake of fruit and vegetables. In the group of foods that possess health risks, food with flavor enhancers and sweet drinks were dominant among other types of food and across different age groups. For supplements, the consumption rate in adults was quite low (<10%). The most popular supplement was vitamins for children, followed by supplements for aesthetics, such as weight management, and supplements to prevent non-communicable diseases [28]. Herbal/traditional products were quite popular with the main purpose of improving immunity and general health [29].

The overall macronutrient intake of Indonesian adults and senior adults was still under the recommended daily allowance. Approximately 45% of them have a total energy intake of <70% of the RDA. The average intake for groups aged 55+ was 1850 kcal for males and 1580 for females, lower than the RDA of 1924 kcal. The carbohydrate intake on average was 225.2 g, less than the RDA of 260 g [30,31]. Even though data show a lower RDA for carbohydrate intake, the experts argued the public consumption of carbohydrates was high due to the Indonesian habit of eating heavy food in the morning, afternoon, and evening with a lot of carbs and having snacks containing mostly carbs. For protein, Indonesian adults and senior adults have a less sufficient protein intake, which leads to increased risk of falls and reduces immune function. Data also indicate that the proportion of people having excess fat intake (>67 g/day) reduces in the senior adult group (17.1%), which is lower than the teenage (30.3%) and adult (28.1%) groups [31]. Nevertheless, experts pointed out that Indonesian adults have certain habits that are difficult to change because they have lived their entire life with those habits, such as consuming fried foods, which could increase the saturated fat.

The micronutrient intake was not much different. There is moderate to high inadequacy of vitamin A, C, and E due to low fruit and vegetable intake leading to high free-radical and low immune function [32]. The folate deficiency is high in adults and senior adults potentially increasing the risk of anemia and cardiovascular disease [33,34,35]. There is also a high deficiency of vitamin B 12 that is important to improve cognitive function and lower the risk of cardiovascular disease [34,35,36]. Vitamin D is also insufficient due to the avoidance of sun exposure, which increases the risk of osteoporosis and reduces muscle strength [37,38]. Even though Indonesia is a tropical country with abundant sunshine, many people avoid the sunlight because it is too hot. Besides vitamins, minerals were also inadequate. Zinc, calcium, and iron were all insufficient and could be associated with the risk of low immune status, cognitive function, anemia, and osteoporosis [32,39,40,41,42]. However, one mineral exceeded the daily recommended allowance—sodium [31]. This might be linked with the habit of eating salty food. High intakes of sodium have been associated with increased high blood pressure leading to the risk of developing heart and kidney problems. A review from one publication also mentioned that appropriate nutritional intervention to achieve healthy aging is mandatory in the adult population as Indonesia expects a significant increase in the elderly population [43].

With increasing recognition of the role of water and adequate hydration in the prevention [44,45] and management of NCDs [46,47], it is essential to have a look at the total water and fluid intake of the Indonesian adult population to have a holistic approach toward ensuring good preparation for having a healthy aging population in the future. Current evidence suggests that dehydration can cause constipation, impaired cognitive function, falling, orthostatic hypotension, salivary dysfunction, and poor control of hyperglycemia in diabetes or hyperthermia [48,49,50,51].

Because of its numerous functions in the human body, water is also an essential nutrient in every life stage. However, with increased age, the body’s water balance mechanisms are at a higher risk of being disturbed due to the aging process. For example, aged kidneys are less able to concentrate urine and retain water during water deprivation [52]. Age-related lower kidney responsiveness to the Anti-Diuretic Hormone (ADH) is thought to play an important role in the loss of renal function [53]. In addition, aging kidneys have a lower ability to regulate sodium excretion [54] adequately. Thus, in older individuals, aged-related physiological changes occur, making the body less able to maintain water homeostasis [55]. Additionally, with an increase in the percentage of body fat (a tissue poor in water) and a decrease of lean body mass due to the aging process, the total body water content will also decrease [53].

In Indonesia, adequacy of water intake started to be explored in 2012 [56], and then, followed by the existence of age- and sex-specific adequate intake (AI) of water recommended by the Indonesian Ministry of Health (2013) [57], and then, this was further detailed and emphasized in Indonesia’s food-based dietary guidelines *Tumpeng Gizi Seimbang* (TGS, Balanced Nutritional Pyramid) [26].

From a national survey in Indonesia in 2016, it was found that 3 out of 10 Indonesians aged 18–65 years old have inadequate intake levels of water from fluids compared to the age- and sex-specific adequate intake (AI) recommended by the Indonesian Ministry of Health (2013) [58]. This proportion is lower compared to previous surveys (55–70%) [59,60]. This improvement could have resulted from some emphasis on promoting the drinking of water in Indonesia. However, further effort is still needed to ensure a decrease of the ID adult population who have inadequate total daily water intake, including the regular evaluation of the impact on things that are already done.

#### 3.2.3. Health Profile

The government of Indonesia launched the Healthy Life Society Movement (GERMAS) to raise public awareness to live a healthy lifestyle, mainly to improve physical activity, to eat healthy food, and to do regular health check-ups. This lifestyle guideline is supported by inter-ministerial coordination, involving the Ministry of Health, Ministry of Agriculture, Ministry of Home Affairs, Ministry of Youth and Sport, Ministry of Education, Ministry of Fisheries, Ministry of Religious Affairs, Food and Drug Administration, and the National Health Insurance Agency. In general, the lifestyle guidelines include promoting physical activities, eating fruit and vegetables, avoiding smoking and alcohol consumption, routine health screening, having a healthy environment, clean water and sanitation. For senior adults, the government recommends routine health monitoring in the integrated guidance post (POSBINDU), developing hobbies based on their health conditions, having balanced nutrition consumption, avoiding smoking, performing elderly exercises, increasing social activity, and managing stress well [61]. The proportion of people with sufficient physical activities is decreasing across the years in all age groups, that is, from 73.9% (2013) to 66.5% (2018) [23]. Indonesian adults have low physical activity as they live sedentary lifestyles. People prefer to use transportation than walking to other places, especially those living in urban cities.

As result of insufficient physical activities and poor eating habits, the prevalence of non-communicable diseases (NCDs) is generally increasing over the years. Overall, the proportion of people with illnesses and health complaints increases with aging. About 51.1% of people aged above 60 had sickness and 26.2% had health complaints [62]. The nutrition-related NCDs, that is, hypertension, diabetes, and heart disease rank as the top three diseases/risks in general adults [23]. In line with this, diabetes and cardiovascular disease also contribute the most to the burden on national health insurance. About 51.6% of the total claims for catastrophic diseases originate from heart disease. Among the Indonesian population in the age group above 15 year-olds, in 2013, hypertension affected 42.1 million of the population (a prevalence of 25.8%), whereas diabetes affected 9 million of the population (a prevalence of 6.9%) [63]. This is with the note that only 30% of both hypertension and diabetes are detected/diagnosed [22]. Both prevalences were found to be increased for hypertension and diabetes by around 20% and triple (300%), respectively in 2018 [23]. According to the 2014 Indonesian Sample Registration System, the most common diseases were stroke (21.1%), heart disease (12.9%), diabetes mellitus (6.7%), tuberculosis (5.7%), and complications of high blood pressure (5.3%) [64].

Not only an economic burden, NCDs also contribute to two-thirds of the mortality in Indonesia. In the midst of the COVID-19 pandemic situation, the presence of NCDs as a comorbid is associated with a high mortality rate [65].

Besides nutrition-related NCDs, adults and senior adults commonly complain about joint problems, mental health problems, insomnia and depression, muscle ache, and being easily tired [23]. Late-onset diseases, for example, osteoporosis, sarcopenia, frailty, and dementia are increasing over the years. Data from the Ministry of Health (2005) showed that the national prevalence of osteoporosis was 10.3% and osteopenia was 41.7% [66,67,68]. The prevalence of sarcopenia among senior adults was 9.1%, and it increases with aging [69,70]. The prevalence of frailty is more than 24%, and the percentage of pre-frail is also high, that is, 61.6% [71,72,73]. While the proportion of dementia was 20.1% as reported in Yogyakarta [34,74].

Apart from the above, in the case of environmental factors, worsening air quality worldwide, including in Indonesia, may also impact the worsening of the health profile via enhancing chronic inflammation status. Systemic inflammation has been recognized as an underlying mechanism for many chronic diseases, including cancer and cardiovascular disease.

Inflammation induced by PM with an aerodynamic diameter of fewer than 2.5 μm (PM2.5) is hypothesized to be a biological link between air pollution and increased morbidity and mortality of chronic diseases, especially cardiovascular disease [75,76]. Worldwide, 9 out of 10 people were found to breathe polluted air, and it is estimated that 7 million people die annually due to air pollution [77]. A study in 2020 on 16 large Indonesian cities showed that the majority exceeded Indonesian annual ambient quality standards, owing primarily to traffic emission and biomass burning. Additionally, volcanic emissions and forest and peat fires have also affected Indonesia’s cities [78]. Unfortunately, there is limited study at the national level on understanding the impact of air pollution on the NCDs prevalence among adult populations in Indonesia.

Over the last two decades, from 1998 to 2016, Indonesia went from being one of the cleaner countries in the world to one of the top 20 most polluted due to a 171% increase in particulate air pollution [79].

Although Indonesia does not currently have a PM2.5 standard that all regions are expected to meet, the government has begun to take initial steps to confront the growing particulate pollution problem [79].

In 2017, the Indonesian government required that all gasoline-fueled vehicles adopt Euro-4 fuel standards by September 2018 (an internationally recognized fuel standard initially adopted in the European Union and now widespread worldwide). Euro-4 demands the use of high-quality, cleaner fuels with a sulfur content less than 50 parts per million (ppm)—10 times more stringent than the sulfur limit in the Euro-2 fuel that Indonesia previously used [79].

The government has also stepped up its efforts to combat air pollution from peat and forest fires. After the 2015 Southeast Asian Haze caused international health and economic damages, Indonesian President Joko Widodo enacted a moratorium on new peatland development and established the Peatland Restoration Agency (BRG). The BRG’s efforts to rewet degraded peatlands have been cited as one possible reason why Indonesia has since experienced fewer fires [79].

Central obesity, smoking, and low physical activities are the main risk factors behind NCDs. Indonesia is among the top 10 countries with the highest smoking rate, with 35% of the population being smokers, consuming on average 12.8 cigarettes per day [23]. The proportion of central obesity is also quite high among adults and senior adults, peaking at the age of 45 to 54 years old [23]. Consumption of fatty foods and low intake of fruit and vegetables are the main risk factors for central obesity. Despite the high prevalence of obesity in the general population, there is a risk of being underweight, especially in senior adults [23]. They might have lower nutrition absorption function and less appetite, resulting in higher underweight prevalence compared to other age groups.

## 4. Strength and Limitation

The purpose of this paper is to document the nutritional status, socio-demographic, and health profile of the adult and senior adult population to prepare for healthy aging in Indonesia. It is beyond the scope to look deeper into the causes of the identified problems. Data used and presented in this paper are based on national surveys from legitimate bodies. Despite the vast amount of literature used in this paper, we did not analyze the quality of the studies. Additionally, the findings presented in this study relied heavily on studies published by the government. Nevertheless, this paper has provided the national picture of the health and nutrition situation of senior adults in Indonesia. It can be used as baseline data to identify the problems and monitor the changes over time.

## 5. Conclusions and Recommendation

Population aging is inevitable: we will have more elderly than children, and people live longer than ever before. The highest population growth in Indonesia is currently found among the age group above 60. This study summarizes the overall demographic status and health profile of adults in Indonesia. The adults have a better education level, literacy rate, and the highest rate in accessing technology. Nevertheless, they have poor lifestyle habits, for example, smoking, and low physical activities. There is also unbalanced nutrition intake in terms of macronutrients and micronutrients among adults. NCDs are the most prevalent diseases in adults, particularly cardiovascular disease and diabetes in terms of health profile. Acknowledging the profile of the adult and senior adult population in Indonesia, the government has taken some measures to support healthy lifestyles among adults. However, a more aggressive approach is needed, especially in converging the existing efforts starting from planning to evaluation, which includes multi-sector action, from government and non-government parties, that is, academic institutions, private companies, and non-governmental organizations. This should be carried out to prepare Indonesia for healthy aging and facilitate the turning of things that are known into actions that meet the local cultural norms of the areas that are approached.

## Figures and Tables

**Table 1 nutrients-13-03441-t001:** Summary of the key issues on socio-demographic status, nutritional status, nutrients intake and health profile of the adult population in Indonesia.

Socio-Demographic Status	Nutritional Status	Nutrients Intake	Health Profile
Increase old-dependency ratio Increase number of elderly Increase literacy rate High access to internet High inequality (Upper-and middle-class vs low-class) High social activity (family & volunteering events)	Increase prevalence of overweight in 2013–2018 Increase prevalence of central obesity in 2013–2018	Low intake on fruits, vegetables, and animal products Protein and fibers deficiency Excess sugar and fat Folate and vitamin D deficiency Zinc, calcium, iron deficiency Excess sodium	NCD (hypertension, diabetes, heart disease) is the most prevalent Low physical activities High prevalence of central obesity Smoking habits Increase of air pollution

## Data Availability

Not applicable.

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
