# Peer review of "The Road to Healthy Ageing: What Has Indonesia Achieved So Far?"

_nutrients, 2021, doi:10.3390/nu13103441_

Round 1

Reviewer 1 Report

This is about a relatively well-written review of the data regarding the quality of Indonesian health status. To get a more complete picture I would suggest adding information about three points: 1. The effect of water consumption as an important component of the diet in health ageing 2. The effect of air pollutants exposure in inflammation status 3. As the leading causes of death worldwide are cardiovascular diseases followed by cancer it will be suggested to add information about state campaigns aimed at reducing the incidence of these diseases

Reviewer 2 Report

As you may have noticed this research was supported by a nutrition supplementation company, which makes a bit uncomfortable .

The article is very relevant globally! 

The authors should start from the beginning to focus on nutrition related issues and how it relates to healthy aging since that is the title of the article. The text includes this information but a bit too late for the reader. I think the connection of healthy aging to nutrition should be presented up front.

In addition , there are way too many places where the "medical" terminology is not accurate in my view. I tried to highlight in yellow places where the terminology is not correct in English or the sentence as a whole.

This article would be a great example of what can be done from the top down in terms of guidelines to enhance active and healthy aging. How to turn guidelines into actions that meet cultural norms is a real challenge and should be somehow mentioned ? thanks
